# Separate Hydrolysis and Fermentation of Kitchen Waste Residues Using Multi-Enzyme Preparation from *Aspergillus niger* P-19 for the Production of Biofertilizer Formulations

**Apurav Sharma [1], Sakshi Dogra [1], Bishakha Thakur [1], Jyoti Yadav [1], Raman Soni [2] and Sanjeev Kumar Soni [1,*]**

[1] Department of Microbiology, Panjab University, Chandigarh 160014, India; akashapurav5@gmail.com (A.S.); d.sakshidogra@gmail.com (S.D.); bishakhathakur1998@gmail.com (B.T.); jyotikakrala15051998@gmail.com (J.Y.)

[2] Department of Biotechnology, D.A.V. College, Chandigarh 160011, India; ramansoni@yahoo.com

[*] Correspondence: sonisk@pu.ac.in

**Abstract:** This study addresses the management of kitchen waste by transforming it into biofertilizer formulations, utilizing an effective, in-house-developed multi-enzyme preparation. An approach consisting of separate hydrolysis and fermentation bioprocessing processes was used, employing a multi-enzyme preparation from *Aspergillus niger* P-19 to separately hydrolyze kitchen waste, followed by the fermentation of the hydrolysate for the growth of *Klebsiella pneumoniae* AP-407, which has biofertilizer traits. This has led to the simultaneous generation of liquid as well as carrier-based biofertilizer formulations with viable cell counts of $3.00 \times 10^{12}$ CFU/mL and $3.00 \times 10^{12}$ CFU/g, respectively. Both biofertilizer formulations significantly enhanced the morphometric characteristics and leaf chlorophyll contents of *Tagetes erecta*, in addition to enriching the soil with essential nutrients. The current study adopted a novel processing technology for the manufacturing of both carrier and liquid biofertilizers, adopting a zero-waste approach for the management of kitchen waste.

**Keywords:** kitchen waste; enzyme; enzymatic hydrolysis; biofertilizer

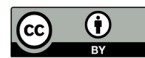

## 1. Introduction

Biodegradable solid waste, particularly kitchen waste and agricultural waste, is a growing global concern due to population growth, changes in consumer behavior, and increased industrial and agricultural activities. The accumulation of organic waste, which is composed of heterogenous polysaccharides including cellulose, hemicellulose, pectin, and starch, not only poses significant environmental challenges but also contributes to pollution, soil erosion, and greenhouse gas emissions [1]. The challenges associated with the management of biodegradable solid waste comprising kitchen waste, which appears to be a very good feedstock for the growth of microorganisms owing to the presence of a variety of nutrients, have gained widespread recognition. As a result, there has been growing interest in researching the development of biopolymers, biofuels, and other value-added products from these waste streams for which all polysaccharides need to be broken down, and this is possible only if we use a mixture of multiple enzyme systems. However, there has been comparatively little attention paid to the potential for producing biofertilizers from biodegradable solid waste. This oversight contrasts with the focus on producing other value-added products, including biofertilizers, to provide significant benefits for sustainable agriculture and soil health. Therefore, exploring the potential for

producing biofertilizers from kitchen waste residues is a crucial area for future research and development [2].

To tackle this problem, there has been a growing emphasis on shifting toward a circular economy as a key priority. The "zero waste approach" advocates for sustainable, long-term socio-economic and environmental benefits by minimizing waste generation through practices such as reusing, repairing, refurbishing, and recycling materials [3]. This method can help mitigate the adverse effects of kitchen waste on the environment and promote the development of a more sustainable society. There has been a marked increase in the identification of plant-growth-promoting bacteria (PGPB), which have been shown to significantly enhance plant growth [4,5]. Using biodegradable solid waste to create biofertilizers with PGPB can reduce the amount of waste sent to landfills and provide a more sustainable and efficient method of fertilizing crops.

Expanding upon our previous research [6], the goal of the current study is to address possible limitations that may arise during the production and isolation of biocompatible organisms in consolidated bioprocessing. Specifically, we aim to investigate the potential of utilizing the organic fraction of solid waste as a feedstock for the production of liquid biofertilizers, which is achieved through the separate depolymerization of the organic residue to release sugars. Additionally, we propose using the solid residue remaining from enzymatic hydrolysis to create effective support for the preparation of carrier-based biofertilizers. This method provides a sustainable solution for managing solid waste residues and producing valuable biofertilizers, which helps to reduce the environmental impact of current waste disposal practices and advance the circular economy.

Consolidated bioprocessing (CBP) is a promising strategy for biofertilizer production as it combines enzyme production, saccharification, and fermentation into a single process, as disclosed by Sharma et al. [6]. However, the need for two biocompatible microorganisms can limit CBP's industrial viability. To develop a more sustainable and industrially viable process, the present study provides an alternative to finding much better microorganisms: one which is capable of producing multiple carbohydrases and another that can use the released sugars separately, with various plant-growth-promoting traits. To achieve this, an approach of separate hydrolysis and fermentation (SHF) is employed, as simultaneous saccharification and fermentation (SSF) may not provide optimal conditions for both enzymes and microorganisms. This is due to differences in the temperature and pH optima between enzymes and microorganisms, which can decrease efficiency and lower product yield. Additionally, the production of enzyme inhibitors by the cultivated microorganism can decrease enzyme efficiency [7–9]. Therefore, SHF is preferred for the development of sustainable and industrially viable biofertilizer formulations.

A novel separate hydrolysis and fermentation bioprocess was developed using an in-house-produced multi-enzyme preparation from *Aspergillus niger* P-19 to disintegrate various polysaccharides in kitchen waste residues. This process resulted in the release of simple sugars and amino acids that support the growth of natural variants of *Klebsiella pneumoniae* AP-407, which can perform atmospheric N-fixation, mobilize P and K, and produce plant-growth-promoting hormones. The bioprocess generated a liquid supernatant and a solid residue, which were used to produce liquid and carrier biofertilizers by cultivating *Klebsiella pneumoniae* AP-407 in the liquid hydrolysate and separating the solid residue, respectively. The industrial application of this process can significantly reduce the cost of nutrient preparation for various biofertilizer formulations and provide a sustainable solution for managing kitchen waste. This process can also minimize dependence on synthetic chemical fertilizers, leading to a more eco-friendly approach.

Building on previous research, our current study introduces an innovative method for converting kitchen waste into biofertilizer formulations using a combination of natural bacterial and fungal strains. This process involves breaking down the complex polysaccharides in the waste using a multi-enzyme system consisting of cellulases, hemi-

cellulases, pectinase, and amylases from a fungal strain, resulting in the production of simple sugars. These sugars are then utilized by a bacterial strain with plant-growth-promoting properties, which ultimately leads to the production of biofertilizers. This approach offers a sustainable solution for waste management and biofertilizer production, utilizing natural microorganisms and reducing reliance on synthetic chemical fertilizers.

## 2. Materials and Methods

### 2.1. Microorganisms

The fungal strain *Aspergillus niger* P-19, already isolated from the natural diversity of Chandigarh city and capable of producing multiple carbohydrases, including a complete cellulase system, hemicellulases, pectinase, and amylases with a compatible temperature and pH optima at 50 °C and pH 4.5, respectively, was chosen [10]. The bacterial strain *Klebsiella pneumoniae* AP-407, which was also already isolated from the natural biodiversity in the rhizospheric soil of Panjab University, Chandigarh, India, was selected for its nitrogen fixation, HCN production, phosphate solubilization, potassium mobilization, siderophore production, ammonia and IAA production abilities [6]. The pathogenicities of the strains were assessed by inoculating them onto blood agar base (HiMedia, Mumbai, Maharashtra, India) plates supplemented with 5% *v/v* sterile, defibrinated sheep blood. Subsequently, the plates were examined for the presence of beta-hemolysis (clear zones), alpha-hemolysis (green zones), and gamma-hemolysis (the absence of clear zones around colonies) after 96 h of incubation [11].

### 2.2. In-House Production of Multi-Enzyme Preparation

The composite kitchen waste used in this study was procured from the hostel and messes of Panjab University, Chandigarh, India. The multi-enzyme system containing cellulases, hemicellulases, amylases, and pectinase was produced in-house, employing the solid-state fermentation of composite kitchen waste with *Aspergillus niger* P-19 in enamel-coated metallic trays with dimensions of 70 cm (L) × 40 cm (B) × 6.5 cm (H). One kilogram of waste was obtained after crushing three kilograms of waste in a blender and squeezing the excess water through a muslin cloth. The resulting waste was dispensed onto a tray, autoclaved, and inoculated with 100 mL of a spore suspension of *Aspergillus niger* P-19 grown on potato dextrose agar (PDA) plates containing $1 \times 10^8$ spores/mL. The mixture was then incubated for four days under stationary state conditions at 25 °C. The moldy waste was then dispensed in 10 L of distilled water, and the enzymes were extracted by blending the contents and filtering them through a nylon sieve. The mycelia-free supernatant, obtained after the centrifugation of the filtered extract at 5000 rpm and 4 °C for 20 min, was analyzed for various enzyme systems, including cellulases, hemicellulases, pectinases, and amylases at 50 °C and a pH of 4.5. The enzymes activities tested using standard procedures included CMCase, FPase, and β-glucosidase (for cellulases) [12]; xylanase [13] and mannanase [14] (for hemicellulases); pectinase [15]; α-amylase [16], and glucoamylase [17] (for amylases). CMCase makes short-chain oligomers containing non-reducing and reducing tails by randomly cutting the amorphous structure of cellulose. FPase produces non-reducing endings that are hydrolyzed to produce cellobiose, a repetitive unit containing two glucose molecules. β-glucosidase hydrolyzes cellobiose units, generating monomeric glucose units. Xylanases randomly split the xylan chain by hydrolyzing glycosidic linkages to release linear and branching oligosaccharides and monomeric xylose units. Mannanases randomly cleave the mannan's β-1,4-linkage internal links, generating new chain endpoints and ultimately releasing mannose sugar moieties from the non-reducing ends of mannan and mannooligosaccharides. Pectinases hydrolyze pectic polysaccharides into monomeric galacturonic acids. α-amylase cleaves the α-1,4-bonds present in the inner regions of amylose and amylopectin to break into oligosaccharides and dextrins, whereas glucoamylase func-

tions as a debranching enzyme by cleaving the final $\alpha$-1,4 links at the non-reducing end of amylase and amylopectin, which releases glucose. The enzyme activities were expressed in terms of International Units (IU)/mL in which one unit of each of the enzymes (CMCase, FPase, β-glucosidase, xylanase, mannanase, pectinase, and glucoamylase) was equal to the amount of enzyme that released one μmole of end product per minute. One unit of $\alpha$-amylase was defined as the amount of enzyme that reduces the color of the starch–iodine complex by 10% in 10 min [6,10].

### 2.3. Partial Purification of the Multi-Enzyme Preparation after Extraction from Solid-State Culture of Aspergillus niger P-19

The crude enzyme preparation was subjected to a two-stage filtration process for partial purification. It was initially passed through a 5-micron polypropylene filter to remove the remaining sediments and dust, followed by another filtration through a 20 kDa membrane.

### 2.4. Enzymatic Hydrolysis of Composite Kitchen Waste Using In-House-Produced Multi-Enzyme Preparation from Aspergillus niger P-19

To prepare the kitchen waste for enzymatic hydrolysis, 250 g of waste was blended and placed in a 2000 mL flask containing 1000 mL of distilled water. The mixture was then autoclaved at 121 °C for 30 min to undergo steam pretreatment. After cooling, 25 mL of the partially purified multi-enzyme preparation from *Aspergillus niger* P-19 was added to the flask. The enzyme preparation had the following activity levels: 6.35 IU/mL of carboxymethyl hydrolyzing (CMCase) activity, 2.15 IU/mL of filter paper hydrolyzing (FPase) activity, 5.80 IU/mL of β-glucosidase activity, 40.85 IU/mL of xylan hydrolyzing (xylanase) activity, 8.25 IU/mL of mannan hydrolyzing (mannanase) activity, 520.16 U/mL of starch liquefying ability ($\alpha$-amylase), 9.00 IU/mL of starch saccharifying ability (glucoamylase), and 8.50 IU/mL of pectin degrading (pectinase) activity. The enzymatic hydrolysis process was carried out by placing the flask in a water bath shaker set at 50 °C and 150 rpm for 48 h.

### 2.5. Fermentation of Enzymatic Hydrolysate of Composite Kitchen Waste for Transformation into Biofertilizer Formulations

The enzymatic hydrolysate was filtered and placed in a flask with a pH of 7.00 ± 0.5. Then, a 10% *v/v* broth culture of *Klebsiella pneumoniae* AP-407 grown overnight was added to the mixture. This culture had a viable cell count of $1.00 \times 10^7$ CFU/mL and possessed various plant-growth-promoting traits such as N-fixation, P solubilization, K mobilization, and the production of plant-growth-promoting hormones. The flask was then incubated in a shaker incubator at 37 °C and 200 rpm for 72 h. After 24 h intervals, samples were taken and analyzed for residual reducing sugars using the DNSA method [18] and glucose using the glucose oxidase–peroxidase method [19] after centrifugation at 10,000 rpm for 10 min at 4 °C. Changes in the viable cell counts of biofertilizer microorganisms were determined using the standard method described by James [20].

### 2.6. Separation of Carrier and Liquid Biofertilizers

After 72 h of incubation, the contents were filtered through a 200-micron double-meshed sieve, and the same process yielded a liquid biofertilizer in the form of the filtrate and a carrier-based biofertilizer after squeezing the solid residue. The final counts of the biofertilizer organisms in the two formulations were also determined and expressed in terms of CFU/mL and CFU/g for the liquid and carrier-based formulations, respectively, and they were stored in polypropylene bottles and air-tight polythene bags, respectively, until further use. Their shelf lives were studied for up to 1 year by observing the residual cell counts at regular intervals of 2 months. The current optimized pro-

cessing technology for transforming kitchen waste into liquid and carrier-based biofertilizer formulations is depicted in Figure 1.

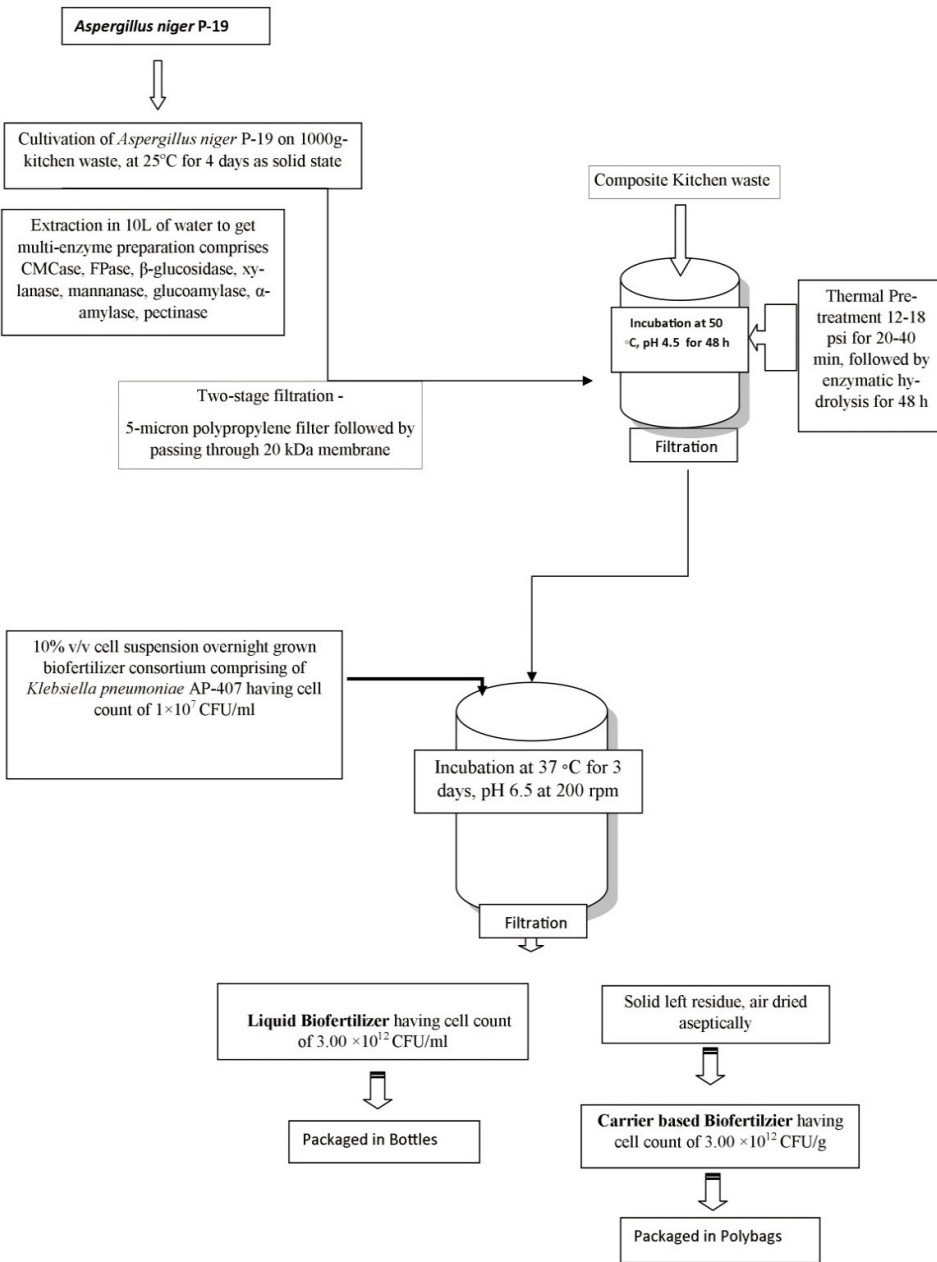

**Figure 1.** Flow diagram revealing various steps involved in the transformation of composite kitchen waste into both carrier and liquid biofertilizer formulations.

*2.7. Seed Germination Test for the Evaluation of Biofertilizer Formulations*

The present study assessed seed germination (SG) and relative seed germination (RSG) using Equations (1) and (2), as described by Luo et al. [21]. In addition, the in vitro seed germination test, or vigor index, was analyzed using Equation (3), according to Jagadeesan et al. [22]. For the experiment, 20 marigold seeds were homogenously soaked in 10% (*w/v* and *v/v*) liquid and carrier biofertilizers, respectively, and triplicate sets were prepared. Following a 1 h soaking period, the seeds were moved onto sterile Petri plates

that contained pre-wetted cotton with sterile, double-distilled water. These plates were then incubated at a temperature of 30 °C for 6 days. After this incubation period, the resulting seedlings were analyzed for their vigor index, seed germination (SG), and relative seed germination (RSG) values using the Equations provided below:

$$SG\ (\%) = \frac{Number\ of\ germinated\ seeds \times 100}{Total\ number\ of\ seeds} \tag{1}$$

$$RSG\ (\%) = \frac{Number\ of\ germinated\ seeds\ (Treated) \times 100}{Number\ of\ germinated\ seeds\ (control)} \tag{2}$$

$$Vigor\ Index = Seed\ germination\ (\%) \times Seedling\ length\ (Root\ length + Shoot\ length) \tag{3}$$

### 2.8. Plant Growth Experiment for the Evaluation of Biofertilizer Formulations

An experiment was conducted to evaluate the impacts of carrier and liquid biofertilizer formulations on the growth of *Tagetes erecta* (marigold) plants. The experiment was carried out from January 2023 to mid-March 2023 at the Department of Microbiology, South Campus, Panjab University, Chandigarh. For each set, 20 seeds were surface-sterilized with 70% ethanol and rinsed three times with sterile distilled water. After shade-drying the seeds for 30 min, they were planted in separate pots with diameters of 28 cm and depths of 20 cm that were each filled with 2500 g of soil, which was sterilized via autoclaving at 15 psi for 1 h. The soil was treated with 2 g of carrier-based biofertilizer or 2 mL of liquid-based biofertilizer, which were mixed with the soil. The pots were then left for 2 h before inoculating the seeds. The same treatment was repeated on the 25th day after taking soil and plant samples. For each treatment, three replicate pots were maintained with a natural photoperiod of 12 h and were watered with tap water for 45 days. After 25, 50, and 75 days of planting and at maturity, three replicates of each treatment were harvested, and various factors were assessed. Morphometric analyses of the host plant for the different treatments were evaluated after 25, 50, and 75 days of planting and at maturity. The morphometric characters evaluated included plant height (cm), shoot height (cm), root length (cm), number of flowers, flower diameter (cm), and flower weight (g). The relative increase in yield for each morphometric character was calculated using the following Equation (4).

$$Relative\ yield\ (\%) = \frac{Yield\ of\ treated\ plant \times 100}{Yield\ of\ control\ plant} \tag{4}$$

### 2.9. Determination of Chlorophyll

The chlorophyll contents of the leaves were measured over 45 days. To do this, one gram of finely chopped, fresh leaves was mixed with 20 mL of 80% acetone. The solution was centrifuged for 5 min (10000 rpm; 4 °C), and the supernatant was collected. The process was repeated until the residue was colorless. The absorbance of the resulting solution was measured at 645 nm and 663 nm against acetone. The concentrations of total chlorophyll, chlorophyll a, and chlorophyll b were determined using the equations described by Arnon [23]:

Total Chlorophyll: $20.2(A645) + 8.02(A663)$
Chlorophyll a: $12.7(A663)–2.69(A645)$
Chlorophyll b: $22.9(A645)–4.68(A663)$

### 2.10. Quantitative Analysis of Soil

The soil was analyzed using a macro- and micronutrient testing kit obtained from HiMedia, India, to determine the organic carbon content of the soil as well as the levels of available phosphate ($P_2O_5$), available potassium ($K_2O$), ammoniacal nitrogen ($NH_3$-N), and nitrate nitrogen ($NO_3$-N) in terms of kg per hectare (kg/ha).

## 3. Results and Discussion

Biodegradable waste from kitchens, vegetable and fruit markets, schools, institutions, and society is typically disposed of via open dumping, burning, or landfilling in underdeveloped or developing nations. The effective management of abundant and regularly produced resources is crucial, as noted by Esteban-Lustres et al. [24]. They emphasized the importance of transforming these resources into new and attractive product lines to promote the development of the bioeconomy. The current process method, therefore, employs a potent cocktail of enzymes to efficiently convert the composite kitchen waste into a sugary hydrolysate which was subsequently transformed into novel biofertilizer formulations. In the current scenario, most of the studies involving separate hydrolysis and fermentation processes are concentrated around biofuel and bio-hydrogen production, which are still yet to be scrutinized for industrial viability. In this context, our current studies provide a much more sustainable process for the management of biodegradable solid waste by separate hydrolysis and fermentation. The two microbial strains used in the study, including *Aspergillus niger* P-19, which is capable of producing a multi-enzyme system, and *Klebsiella pneumonia* AP-407, which is has biofertilizer traits, were found to be non-pathogenic based on their inability to produce any zone around their colonies on blood agar plates, as exhibited in Figure 2.

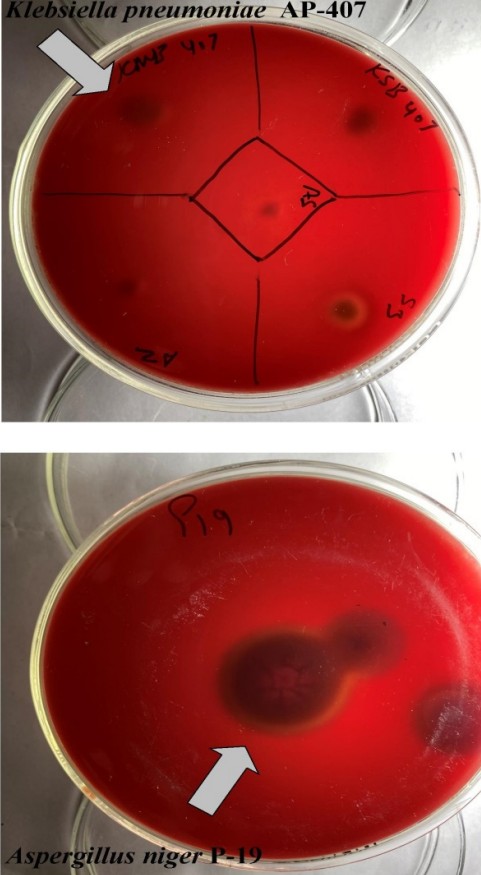

**Figure 2.** Growth of Klebsiella pneumoniae AP-407 and Aspergillus niger P-19 on blood agar plates with no hemolysis.

*3.1. In-House Production of Multi-Enzyme Preparation from Aspergillus niger P-19*

The potential of the fungal strain *Aspergillus niger* P-19 to be used for the production of multiple carbohydrases on an inexpensive substrate such as de-oiled rice bran was already disclosed by Chugh et al. [10]. In a continuation of previous work, the present process involves the separate hydrolysis of kitchen waste to maximize the hydrolysis of kitchen waste so that a maximum amount of sugars can be released. The enzyme activity is expressed in IU/mL, which denotes the quantity of enzyme required to produce 1 μmole of product per min under defined enzyme assay conditions. The crude multi-enzyme preparation obtained from 1000 g of fermented composite kitchen waste (10 L) was subjected to partial purification via two-stage filtration, and the resulting enzyme activities were analyzed. The enzyme activities were found to be as follows: 12 IU/mL of CMCase, 3.15 IU/mL of FPase, 12.80 IU/mL of β-glucosidase, 70.85 IU/mL of xylanase, 20.25 IU/mL of mannanase, 956.16 U/mL of α-amylase, 26.00 IU/mL of glucoamylase, and 19.50 IU/mL of pectinase. *Aspergillus* spp. is known for its potential to produce multiple carbohydrases on various types of biodegradable solid waste and lignocellulosic biomass comprising kitchen waste and deoiled rice bran in solid-state, surface, and submerged fermentation processes [6,10,25,26]. The focus is largely on fungi because of their ability to produce large amounts of hemicellulases and cellulases. Furthermore, the formulation used in the present study consists of a distinct blend of enzymes, encompassing the complete cellulase system in addition to xylanases, mannanases, pectinases, and amylases.

*3.2. Enzymatic Hydrolysis of Composite Kitchen Waste Using In-House-Produced Multi-Enzyme Preparation from Aspergillus niger P-19*

The composite kitchen waste was hydrolyzed via the in-house-produced multi-enzyme preparation. The enzymatic hydrolysis released 31.1 g/L of total reducing sugars and 15.0 ± 0.13 g/L of glucose. The hydrolysate was filtered through a 200-micron double-meshed sieve. A total of 24 ± 1.4 g of remaining solid residue was obtained, which was air-dried overnight, sterilized via autoclaving, and stored in a cold storage facility until further use. The resultant liquid hydrolysate was further sterilized before being employed as a nutrient medium for the growth of microorganisms that create biofertilizers. The enzyme cocktail from *A. niger* P-19 proved to be a source of an effective enzyme cocktail. The enzyme cocktail hydrolyzed the composite kitchen waste, which was then utilized to produce biofertilizer. In our earlier effort to convert kitchen waste into biofertilizers via consolidated bioprocessing, two biocompatible organisms were always needed, one of which could hydrolyze the waste and another which could utilize the sugars generated in the hydrolysate concurrently. To address this issue, we hydrolyzed the kitchen waste separately and obtained the greatest possible amount of sugar, which was more than the amount obtained via consolidated bioprocessing reported earlier [6,27], with 24 ± 1.4 g of solid residue left.

*3.3. Fermentation of Sugars Released after Enzymatic Hydrolysis of Composite Kitchen Waste into Biofertilizer Formulations*

The liquid hydrolysate was effectively converted into liquid biofertilizer using the current method. In our prior study [6], the final viable count of *Klebsiella pneumoniae* AP-407 was found to be $1.03 \times 10^{12}$ CFU/mL. In contrast, in this process, the viable count of *Klebsiella pneumoniae* AP-407 was found to be $3.00 \times 10^{12}$ CFU/mL as depicted in Table 1. The greater sugar release from hydrolysis compared to the sugar release via consolidated bioprocessing is the likely cause of *Klebsiella pneumoniae* AP-407's higher cell count.

**Table 1.** Total reducing sugars, glucose, and microbial cell counts obtained during fermentation on kitchen waste hydrolysate.

| Time (h) | Total Reducing Sugars (%) | Glucose (%) | *Klebsiella pneumoniae* AP-407 (CFU/mL) |
|---|---|---|---|
| 0 | $3.10 \pm 0.155$ | $1.5 \pm 0.075$ | $1.00 \times 10^6$ |
| 24 | $1.80 \pm 0.090$ | 0 | $2.45 \times 10^8$ |
| 48 | $0.75 \pm 0.045$ | 0 | $1.10 \times 10^{10}$ |
| 72 | $0.08 \pm 0.004$ | 0 | $3.00 \times 10^{12}$ |

Columns represent the results of the mean and standard deviation.

### 3.4. Separation of Carrier and Liquid Biofertilizers

Enzymatic hydrolysis was found to be the better hydrolyzing method for the composite kitchen waste as only $24 \pm 1.4$ g of remaining solid residue was obtained from 250 g of composite kitchen waste, which is 9.6% of the total solid mass. The remaining solid residue obtained was further used as the carrier biofertilizer with a healthy viable cell count, thus employing a "zero waste approach". The prepared formulation was air-dried aseptically and packed in air-tight poly bags. The liquid biofertilizer preparation was packed in sterilized bottles until further use. As far as we know, there are no other studies available on the preparation of biofertilizer formulations using consolidated bioprocessing except for the one previously reported from our laboratory [6]. Most of the studies concentrate on the production of biofertilizers using a traditional method in which biofertilizer organisms are usually cultivated on their specific growth media. Even after much exploration of the use of biodegradable solid waste as a production medium for the preparation of biofertilizer, the shortest and least expensive method comes from our laboratory, including the present study.

### 3.5. Physico-Chemical and Biological Characterizations of Developed Biofertilizer Formulations

The carrier and liquid biofertilizers developed from the present processing technology were further analyzed by assessing their various physicochemical and biological characteristics. The prepared biofertilizer formulations carried healthy counts of microbes with plant growth-promoting traits. Overall, the nutrients from the kitchen waste hydrolysate and the plant-growth-promoting traits of biofertilizer microorganisms are a better substitute in comparison to traditional fertilizers and biofertilizers. The characteristics of the kitchen waste hydrolysate and the biofertilizer formulations developed from the kitchen waste hydrolysate using the present processing technology are depicted in Table 2. The kitchen waste hydrolysate itself is enriched with various micronutrients released from hydrolysis. Furthermore, the inclusion of *Klebsiella pneumoniae* AP-407 significantly enhanced the chemical and biological characteristics of the kitchen waste hydrolysate.

**Table 2.** Characteristics of the kitchen waste hydrolysate and biofertilizer formulations.

| Parameter (s) | Kitchen Waste Hydrolysate | Carrier-Based Biofertilizer | Liquid Biofertilizer |
|---|---|---|---|
| pH | $4.0 \pm 0.5$ | $6.5 \pm 0.5$ | $6.5 \pm 0.5$ |
| Viable Count | Nil | $1.00 \times 10^{12}$ CFU/g | $3.00 \times 10^{12}$ CFU/mL |
| IAA | Nil | $31.75 \pm 1.75$ µg/mL | $34.40 \pm 1.60$ µg/mL |
| HCN | Nil | + | + |
| Siderophore | Nil | Hydroxymate (+) | Hydroxymate (+) |

Columns represent the results of the mean and standard deviation.

+ signifies the presence of trait

Biofertilizers with a long shelf life that can be feasibly used and allow for the controlled dispersion of the researched microorganisms are urgently needed in the agro-industrial sector. The potential for the natural, affordable recovery of proteins and carbohydrates from agricultural biomass is enormous [28]. The liquid biofertilizer formulation from the aforementioned processes was used in the current approach, together with the solid residue remaining from the hydrolysis of kitchen waste as a carrier for inoculum adsorption. As it contains significant amounts of carbon and other micronutrients, the remaining kitchen waste solid residue serves as a stabilizing supply of these elements. The colony-forming units (CFU/mL and CFU/g) of the biofertilizer developed in our study were found to be superior to those reported by Xu et al. [29], who achieved a maximum of $9.7 \times 10^9$ CFU/mL while preparing a biofertilizer from sweet potato starch wastewater. Our biofertilizer formulations meet the requirements of the Fertilizer Control Order (India), which mandates a minimum count of $1 \times 10^8$ CFU/mL for liquid biofertilizer or $5 \times 10^7$ CFU/g for powder, granules, or carrier materials after six months [30].

After undergoing separate hydrolysis and fermentation processes, both the liquid and carrier biofertilizers were prepared, packed, sealed, and stored in a cold room. Before storage, the final cell counts for liquid biofertilizer and carrier biofertilizer were $3.00 \times 10^{12}$ CFU/mL and $3.00 \times 10^{12}$ CFU/g, respectively. Following a 10-month storage period, the cell count for the liquid biofertilizer reduced to $2.20 \times 10^7$ CFU/mL, while the cell count for the carrier biofertilizer reduced to $1.10 \times 10^5$ CFU/g, meeting the requirements set by FCO (India) [30]. Studies by Allouzi et al. and Raimi et al. [31,32] support the better shelf life of liquid biofertilizers. These biofertilizers have advantages over solid inoculants, such as an increased resistance to contamination, no requirement for sticky materials, application via modern machinery, the ability to withstand high temperatures up to 45 °C, user-friendliness, and the option to add ingredients that enhance microbial growth [31,33–35].

Altogether, the concept of hydrolyzing kitchen waste separately to enhance hydrolysis for improved sugar production led to an increase in the number of viable cells in liquid biofertilizer formulations, in addition to the creation of a carrier-based biofertilizer.

### 3.6. Influence of Biofertilizer Formulations on Seed Germination

The prepared carrier-based and liquid biofertilizers significantly enhanced the seed germination and relative seed germination of marigold seeds. The liquid-based biofertilizer demonstrated 90.0 ± 3.75% seed germination in comparison to the control set, which showed 60.0 ± 2.25% seed germination, as depicted in Figure 3. The fastest vigor index was observed in the liquid-biofertilizer-treated seeds, followed by the carrier-biofertilizer-treated seeds, as depicted in Table 3. The vigor index in the case of the carrier biofertilizer was 275.90, and the vigor index was 620.00 in the liquid biofertilizer in contrast to the control, which had a vigor index of 185.25. Jagadeesan et al. [22] recently prepared a biofertilizer using chicken feather waste which was enriched with a biofertilizer strain of *Bacillus pumilus.* The present study results overlap with the results of Jagadeesan et al. [22] in terms of the enhancement in the vigor index and seed germination of *Tagetes erecta* (Marigold), as the biofertilizer formulations shortened the growth span of the seeds.

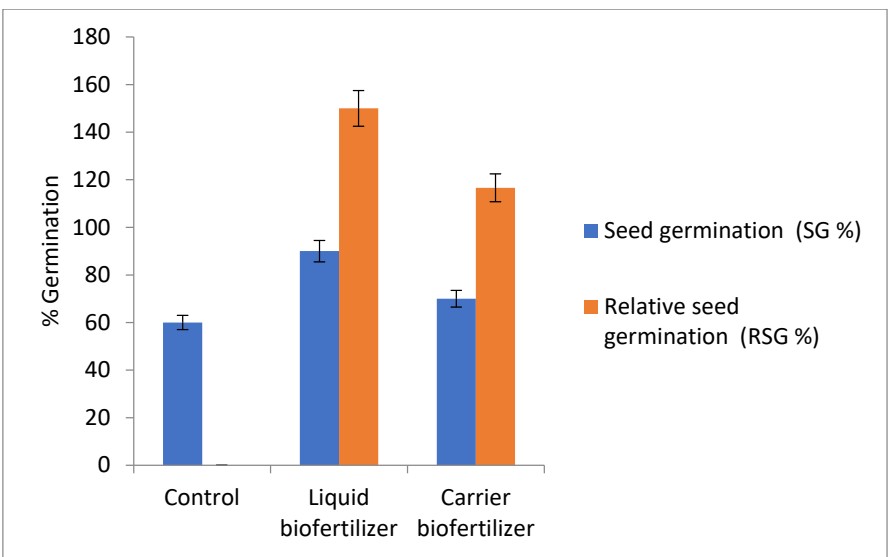

**Figure 3.** Effects of liquid and carrier-based biofertilizers on seed germination (SG) and relative seed germination (RSG) of *Tagetes erecta* (marigold).

**Table 3.** Effects of liquid and carrier-based biofertilizers on the number of flowers, flower diameter (cm), and flower weight (g) of *Tagetes erecta* (marigold) after 25, 50, and 75 days of plant development assay.

| Parameter | Control | Carrier | Liquid | Relative Yield (%) | |
| | | | | Carrier | Liquid |
| --- | --- | --- | --- | --- | --- |
| Number of flowers | 40 ± 2 | 56 ± 3 | 65 ± 3 | 140.0 | 162.5 |
| Flower diameter (cm) | 5.85 ± 0.092 | 6.4 ± 0.150 | 7.0 ± 0.105 | 109.4 | 119.6 |
| Average Flower weight (g) | 6.15 ± 0.236 | 7.4 ± 0.220 | 8.5 ± 0.210 | 120.3 | 138.2 |

Columns represent the results of the mean and standard deviation. All the values differ from the control significantly by the Holm–Sidak test, with $p < 0.001$.

### 3.7. Influence of Biofertilizer Formulations on Plant Development Assays

The experiment showed that both liquid and carrier-based biofertilizers had significant positive effects on the growth and yield of *Tagetes erecta* (marigold). After 75 days, plant height was +12.5 cm more in the plants treated with the carrier-based biofertilizer and +29.0 cm more in the liquid-biofertilizer-treated *Tagetes erecta* (marigold) compared to the control. Similarly, in the liquid- and carrier-biofertilizer-treated *Tagetes erecta* (marigold), the shoot heights were +23.1 cm and +10.1 cm, respectively, compared to the control. Both the liquid and carrier biofertilizers had positive impacts on all morphometric traits of *Tagetes erecta* (marigold), including plant height, shoot height, root length, number of flowers, flower diameter, and flower weight. Figure 4 depicts the morphometric characteristics of the plant sets with and without biofertilizer treatments, showing significant enhancements in the treated plant sets. The relative yield was used to determine the actual increase in yield in each morphometric trait of the plant. The percentage relative increases in yield in the plants treated with carrier-based biofertilizer after 75 days were 133.7%, 137.2%, 123.0%, 140.0%, 109.4%, and 120.3% for plant height, shoot height, root length, number of flowers, flower diameter, and flower weight, respectively. The liquid biofertilizer had a more significant impact on the growth of plants. The percentage relative increases in yield in liquid-biofertilizer-treated plants after 75 days were 178.3%, 185.2%, 158.0%, 162.5%, 119.6%, and 138.2% for plant height, shoot height, root length, number of flowers, flower diameter, and flower weight, respectively. The use of carrier-based biofertilizer treatment and liquid biofertilizer treatment increased the plant height by 33.7%, and 78.3% in the treated plant sets, respectively. The shoot height in-

creases were 33.7%, and 85.2% in carrier-based biofertilizer-treated and liquid biofertilizer-treated plant sets, respectively. A similar trend was observed in the increases in root height, which were 23.0%, and 58.0% in carrier-based-biofertilizer-treated and liquid-biofertilizer-treated plant sets, respectively. The results are summarized in Tables 3 and 4.

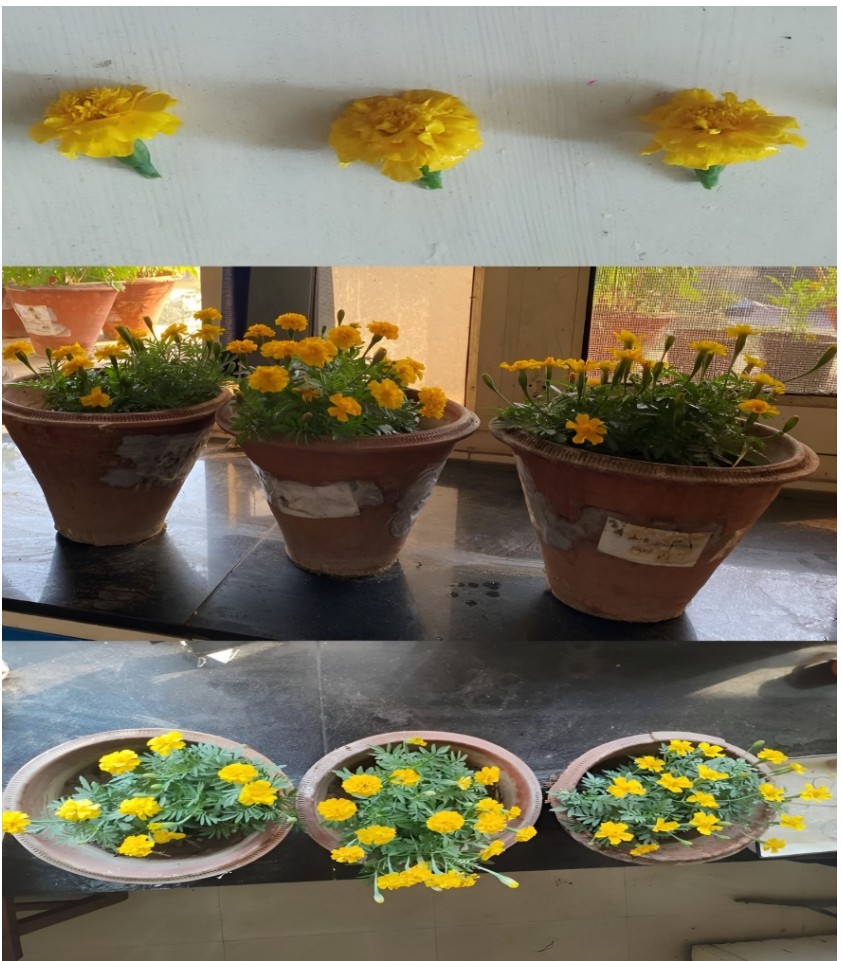

**Figure 4.** Morphometric traits of different treatments on *Tagetes erecta*. Left to Right (**left** pot: control, **middle** pot: liquid biofertilizer, and **right** pot: carrier biofertilizer).

**Table 4.** Effects of liquid and carrier-based biofertilizers on the heights of different parts of *Tagetes erecta* (marigold) during plant development.

| Height (cm) | Day 25 | | | Day 50 | | | Day 75 | | | Relative Height (%) | |
|---|---|---|---|---|---|---|---|---|---|---|---|
| | Control | Carrier | Liquid | Control | Carrier | Liquid | Control | Carrier | Liquid | Carrier | Liquid |
| Plant | 11.7 ± 0.280 | 13.0 ± 0.500 | 15.5 ± 0.650 | 28.5 ± 0.425 | 32.5 ± 0.625 | 47.5 ± 0.875 | 37.0 ± 1.350 | 49.5 ± 1.475 | 66.0 ± 1.800 | 133.7 | 178.3 |
| Shoot | 9.7 ± 0.230 | 10.0 ± 0.350 * | 11.5 ± 0.380 | 21.5 ± 0.250 | 24.0 ± 0.400 | 38.0 ± 0.625 | 27.1 ± 0.905 | 37.2 ± 1.010 | 50.2 ± 1.310 | 137.2 | 185.2 |
| Root | 2.0 ± 0.050 | 3.0 ± 0.150 | 4.0 ± 0.410 | 6.5 ± 0.115 | 8.5 ± 0.220 | 9.5 ± 0.325 | 10.0 ± 0.445 | 12.3 ± 0.465 | 15.8 ± 0.490 | 123.0 | 158.0 |

Columns represent the results of the mean and standard deviation. All the values differ from the control significantly by Holm–Sidak test with $p < 0.001$ except for those marked with *.

The current project successfully transformed biodegradable solid waste into carrier and liquid biofertilizers in an economically viable manner. The carrier and liquid biofertilizers' effects on plant development and soil were consistent with the underlying concept. The use of carrier and liquid biofertilizers made from composite kitchen waste resulted in a significant increase in both plant yield and soil fertility. Over 75 days, the applications of carrier and liquid biofertilizers resulted in significant increases in various plant growth parameters, including plant height, root height, plant fresh weight, number of flowers, flower diameter (cm), and flower weight (g). The enhanced growth and yield of plants can be attributed to several factors, including the presence of indoleacetic acid (IAA), which has been shown to improve plant growth yield in studies by Xu et al., Bhardwaj et al., and Kumar et al. [29,36,37]. Phosphate solubilization and ammonia excretion have also been associated with improved growth [29,38,39]. Additionally, the production of hydrogen cyanide (HCN) and siderophore by plant-growth-promoting rhizobacteria (PGPR) can act as protecting agents for plants under stressful conditions and contribute to an improved yield [40]. El Komy et al. [41] conducted a trial using a combination of *Azotobacter*, *Azospirillum*, and *Klebsiella* strains to manage root rot disease and enhance sunflower growth through nitrogen fixation, phosphate solubilization, and the production of indoleacetic acid (IAA), siderophore, and hydrogen cyanide (HCN). The present study suggests that an increased uptake of nitrogen, phosphorus, and potassium, as well as IAA biosynthesis, ammonia production, siderophore production, and HCN production, may have contributed to the significant growth of Tagetes erecta (marigold) observed. The present results are also supported by our recent study on *Brassica juncea* for 45 days trial and disclose the efficacy of *Klebsiella pneumoniae* AP-407 on plant growth and in improving soil quality. Another study in which a biofertilizer-mediated improvement in plant mineral nutrients was observed was the study by Badawy et al. [42] in which a biofertilizer strain was observed to reduce Cd and Ni in the soil environment, in addition to improving the plant height and increasing the chlorophyll content.

### 3.8. Influence of Biofertilizer Formulations on Chlorophyll Content

The liquid biofertilizer had the greatest influence on the chlorophyll content of *Tagetes erecta* (marigold), resulting in the highest chlorophyll content (a + b) of 83.5 µg/mL, followed by the carrier-based biofertilizer, with a chlorophyll content of 65.25 µg/mL (Figure 5). The control set of *Tagetes erecta* (marigold) had the lowest chlorophyll content (a + b) of 46.8 µg/mL. The increase in chlorophyll content may indicate an improvement in the photosynthetic efficiency of the plants, which can lead to better growth and yield. The improved levels of chlorophyll in liquid- and carrier-biofertilizer-treated plants also emphasizes the potential of the formulations prepared from the present processing technology. The two essential ingredients for the production of chlorophyll are nitrogen and potassium [43], which are attributed to better chlorophyll contents in healthy plants. The higher chlorophyll content is also attributed to the bio-stimulatory impact of the microorganisms present [44]. The carrier and liquid biofertilizers also significantly improved the chlorophyll contents of *Tagetes erecta* (marigold), which can be attributed to the improved availability of nitrogen and phosphorus. The findings of Zafar-ul-Hye et al. [45] support our findings; these authors also found similar results while working with cadmium-resistant rhizobacteria for the availability of nitrogen and phosphorus. The ability of *Klebsiella* sp. GR9 to increase rice output was also highlighted by Govindarajan et al. [46], who attributed it to the GR9 strain's effectiveness in fixing nitrogen.

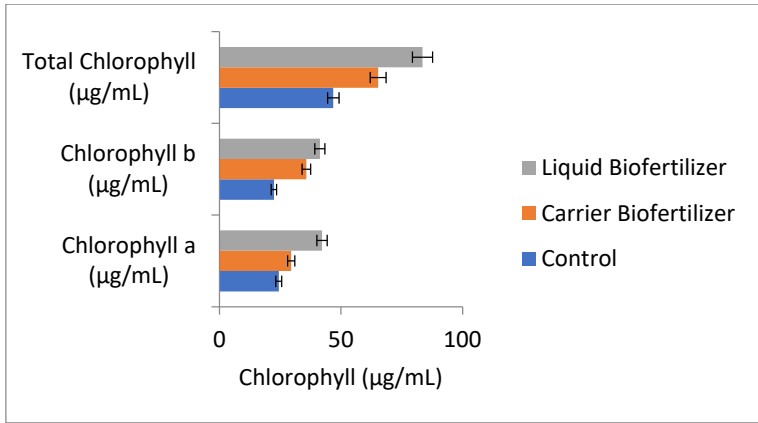

**Figure 5.** Effects of liquid and carrier-based biofertilizers on chlorophyll (chl a, chl b, and chl a + b) contents of *Tagetes erecta* (marigold).

### 3.9. Quantitative Analysis of Soil

The measurement of the available nutrients in the soil, namely, phosphate ($P_2O_5$), potassium ($K_2O$), ammoniacal nitrogen ($NH_3$-N), and nitrate nitrogen ($NO_3$-N), in kilograms per hectare (kg/ha) is depicted in Figure 6. The analysis showed that the amount of available phosphate ranged from 22 to 56 kg/ha, while the available potassium ranged from 112 to 280 kg/ha. On the other hand, the ammoniacal nitrogen level was observed to be low, at approximately 15 kg/ha, and there was no detectable nitrate nitrogen on the day of sowing *Tagetes erecta* (marigold) in the soil. Both the liquid and carrier biofertilizer treatments significantly increased the levels of available phosphate ($P_2O_5$), available potassium ($K_2O$), ammoniacal nitrogen ($NH_3$-N), and nitrate nitrogen ($NO_3$-N) compared to the control group of plants. The synergistic effect of the kitchen waste hydrolysate obtained from enzymatic hydrolysis and the bacterial biofertilizer strain enhanced the macro- and micronutrient levels in the soil. The level of available phosphate in the liquid biofertilizer was the highest on the 25th day, with a level between 70 and 80 kg/ha, and the highest on the 25th day in the carrier, with the same level. In contrast, in the control group, this level peaked on day 25 at below 50 kg/ha. The available potassium levels showed a similar pattern, peaking on the 25th and 50th days, with levels ranging from 350 to 400 kg/ha in both the liquid and carrier biofertilizers. The highest level of potassium was observed on the 25th day in the carrier biofertilizer, with the same level as the liquid biofertilizer. In contrast, the control group reached its peak on day 25, with a significantly lower level of just 110 to 120 kg/ha. The most important nutrient in plant growth is nitrogen. Nitrate nitrogen is only produced in the presence of microbial strains that are capable of fixing nitrogen for plants. The nitrate nitrogen was not observed in the control set of plants, whereas in the liquid-biofertilizer-treated plants, it peaked on the 25th and 50th days, with a level of around 50 kg/ha, and in the case of the carrier-biofertilizer-treated plants, it peaked on the 25th day with the same level, after which it gradually dropped, which could be attributed to the utilization of the nutrients by the plants.

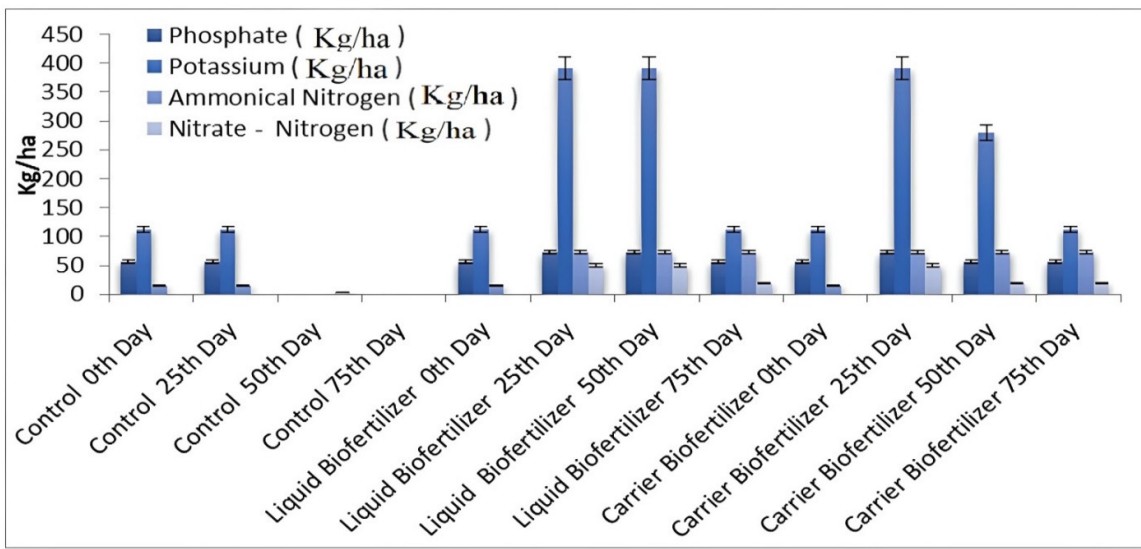

**Figure 6.** Effects of liquid and carrier-based biofertilizers on available phosphate (P₂O₅), available potassium (K₂O), ammoniacal nitrogen (NH₃-N), and nitrate nitrogen (NO₃-N) in the soil during the soil growth experiment.

Overall, the control set had low levels of phosphorus and potassium, whereas the plant sets treated with biofertilizer not only used phosphorus and potassium but also solubilized the phosphate and potassium that were already present in the soil. Figure 6 suggests that even after 25 and 50 days in the soil, the levels of phosphate and potassium were noticeably higher. The minerals in the kitchen waste and the characteristics of *Klebsiella pneumoniae* AP-407 that encourage plant development are responsible for the elevated amounts of phosphorus, potassium, and nitrogen in the biofertilizer-treated plant sets. The higher level of available phosphorus is also attributed to the presence of microorganisms; as in a study by Semerci et al. [47], better phosphorus solubilization was observed from sewage sludge ash in the presence of microorganisms with the capability to dissolve phosphorus. The present study investigated the effect of bacterial inoculation on the growth of bacterial colonies in soil and the subsequent increase in the availability of phosphorus, which is immobilized in the soil as a poorly soluble compound. The study results indicate that bacterial inoculation can increase bacterial colony growth and phosphorus release into the soil solution under favorable conditions. Furthermore, it was observed that bacteria aid in the mineralization of organic matter, including organic or mineral–organic fertilizers, when added to the soil [48]. This suggests that bacterial inoculation could be a valuable technique for improving soil fertility and nutrient availability in agricultural systems.

These findings confirm the biofertilizer's high quality and are corroborated by a study from Xu et al. [29] in which a biofertilizer was formulated utilizing wastewater from sweet potato starch. According to Tiquia [49], the ammonification (NH₄⁺) process, which turns organic nitrogen into NH₃ and NH₄⁺ ions, is the cause of nitrogen loss from the soil. Both the control soil used in the current investigation and the compost made from chicken feathers used by Nagarajan et al. [50] showed similar findings. In contrast, as indicated by Muhammad et al. [51], Sun et al. [52] and the present study, the soil treated with liquid and carrier biofertilizers preserved nitrate (NO₃-N) nitrogen (NH₃-N), which is essential for creating and sustaining a nitrogen pool in the soil. The findings of this study thus provide a method for preserving the soil's nitrogen pool and preventing nitrogen loss from agricultural soil.

Building upon the findings of our previous study [6], in which we implemented an efficient consolidated bioprocessing approach to convert composite kitchen waste into

biofertilizers, in the present work, we have further improved this process by adopting a strategy of separate hydrolysis and fermentation which enhances the sustainability of agro-industrial product production. To assess the effectiveness of this approach, we have compared the biofertilizer formulations developed in this study with those reported in the previous literature that employed different types of agro-industrial wastes. Table 5 presents a comprehensive comparison of the formulations, including their environmental impacts and the quality of their products. Our results demonstrate the significance of this study in advancing the development of sustainable and high-quality soil-nourishing agro-industrial commodities.

We also conducted a comparative analysis with several prominent studies in the field of producing agro-industrial products with soil-nourishing properties. Our findings demonstrate that our proposed technology for producing biofertilizer formulations is more sustainable, industrially viable, and environmentally friendly than existing technologies. Moreover, our method offers a shorter production process and effective waste management. Overall, our study provides a promising alternative for the production of biofertilizer formulations that can enhance soil fertility and promote sustainable agriculture.

**Table 5.** Comparative analysis of present processing technology with other studies involving different agro-industrial wastes transformed into various agro-industrial commodities having soil nourishment traits.

| Agro-Industrial Waste | Process Involved | Microorganism Involved | Agro-Industrial Commodity Generated | Impact | Reference |
|---|---|---|---|---|---|
| Food waste | Food waste inoculated with microbes in a composter at 50 °C for 28 days | *Brevibacillus borstelensis* SH168 | Biofertilizer with $1.82 \times 10^9$ CFU/g | Food waste in addition to biofertilizer production | [53] |
| Wastewater from sweet potato starch | Inoculation in 100 mL of sterilized (121 °C, 20 min) SPSW and incubated at 24–32 h incubation at 30 °C | *Paenibacillus polymyxa* | Biofertilizer with $9.7 \times 10^9$ CFU/mL | Biofertilizer that improves the growth of a tea plant | [29] |
| Peat, corn cobs with 20% (*w/w*) perlite, wheat husks with 20% (*w/w*) perlite, and composted cattle manure with 20% (*w/w*) perlite | Adsorption of *Aspergillus niger* 1107 on a carrier material developed from waste | *Aspergillus niger* 1107 | Phosphate Biofertilizer | Higher growth and high content of phosphate in soil | [54] |
| Fruit waste | Between 30 and 40 days of the composting process | *Bacillus* spp. and *Aspergillus* spp. | Carrier-based biofertilizer | Better seed germination, shoot and root heights, and the ability to prevent root diseases | [55] |
| Sawdust and agricultural waste | Biofertilizer was produced from agro wastes by composting | *Actinomyces* spp., *Streptomyces* spp., and *Rothia* spp. | Biofertilizer (compost) | Better plant height and higher leaf width indicate a higher rate of photosynthesis | [56] |
| Chicken feather waste | A total of 30 days of degradation process using 20–25% inoculum *w/w* | *Bacillus subtilis* | Compost | Management of chicken feather Increases in N, P, and K contents of the soil | [50] |
| Caribbean pine sawdust | An amount of 2.0 g biochar adsorbed with inoculum and | *Pseudomonas* sp., *Serratia* sp., and *Kosakonia* sp. | Biofertilizer with $1.0 \times 10^7$ | Increases seedling growth nutrient in soil and growth of *Allium cepa* L. | [25] |

| | | | | | |
|---|---|---|---|---|---|
| | shaken at 150 RPM, 24 h at 30 ± 2 °C | | | CFU/mL | |
| Chicken feather waste | White chicken feathers inoculated with *B. pumilus* AR57 in 1% *v/v*; 1.25 × 10$^8$ CFU/mL) and incubated at 150 rpm, 37 °C for 28 h | *Bacillus pumilus* AR57 | Biofertilizer | Enhances total phosphate and potassium solubilizers and nitrifying bacteria in the soil of *Zea mays* L. | [22] |
| Kitchen waste | Separate hydrolysis and fermentation for 5 days | *Aspergillus niger* P-19 and *Klebsiella pneumoniae* AP-407 | Carrier and liquid biofertilizer formulations with 3.00 × 10$^{12}$ CFU/g and 3.00 × 10$^{12}$ CFU/mL, respectively | Kitchen waste management in addition to biofertilizer production improves both plant growth of *Tagetes erecta* (Marigold) and soil quality | Present study |

## 4. Conclusions

The current study proposes an attractive alternative for managing kitchen waste by converting it into carrier and liquid biofertilizers. The study effectively validated an in-house-produced multiple enzyme preparation that efficiently hydrolyzed composite kitchen waste. This was followed by fermentation with a biofertilizer strain of *Klebsiella pneumoniae* AP-407 to produce biofertilizer formulations with viable cell counts of $3.00 \times 10^{12}$ CFU/mL and $3.00 \times 10^{12}$ CFU/g for liquid and carrier biofertilizers, respectively. Both biofertilizer formulations significantly enhanced the morphometric characteristics and chlorophyll contents of leaves of *Tagetes erecta*, in addition to enriching the soil with essential nutrients. Biofertilizer formulations enhanced the available nutrients in the soil, namely, phosphate ($P_2O_5$), potassium ($K_2O$), ammoniacal nitrogen ($NH_3$-N), and nitrate nitrogen ($NO_3$-N). These results exceed the claims of any previous study. The approach outlined in this study presents an affordable and efficient solution for solid waste management and biofertilizer production and has the added benefit of extending the shelf life of the final product. If scaled up for commercial use, this strategy has the potential to revolutionize the way we manage municipal solid waste and produce biofertilizers at a low cost, which is especially important given the high demand for such products in the agricultural industry. This could lead to a more sustainable approach to waste management and contribute to the overall improvement of soil health and agricultural productivity.

**Author Contributions:** A.S.: investigation, writing—original draft preparation, writing—review and editing; S.D.: investigation, writing—original draft preparation, writing—review and editing; B.T.: investigation, writing—review, and editing; J.Y.: investigation, writing—review, and editing; R.S.: investigation, writing—original draft preparation, writing—review and editing, supervision; S.K.S.: Conceptualization, writing—review and editing, supervision. All authors have read and agreed to the published version of the manuscript.

**Funding:** This research received no external funding.

**Institutional Review Board Statement:** Not applicable.

**Informed Consent Statement:** Not applicable.

**Data Availability Statement:** Not applicable.

**Conflicts of Interest:** The authors declare no conflicts of interest.

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
