# Peer review of "Separate Hydrolysis and Fermentation of Kitchen Waste Residues Using Multi-Enzyme Preparation from Aspergillus niger P-19 for the Production of Biofertilizer Formulations"

_sustainability, doi:10.3390/su15129182_

Round 1
Reviewer 1 Report
How to deal with kitchen waste considering both efficiency and environmental effect is an important issus in modern society. Bio-method is one of the most popular way. Authors used multi-enzymes & microbes to dispose waste into valued biofertilizer formulations, which provided an new way for waste treatment. But authors should improve quality of figures and organization of table before acceptence.
Author Response
We sincerely appreciate your diligent evaluation and constructive criticism, which have undoubtedly helped us improve the overall quality of the manuscript.

Reviewer 2 Report
This article presented the application of the enzymes prepared from Aspergillus niger P-19 to separately hydrolyze kitchen waste, and the hydrolysis and fermentation bioprocessing were studied. In addition, the growth of Klebsiella pneumoniae AP-407 in the liquid hydrolysate as well as the simultaneous production of carrier-based biofertilizer was achieved.
The experimental work is somewhat good, and the data from this research are acceptable. The topic of this study is filled within the scope of the journal. Before acceptance, I may suggest suitable revisions for this work.
Major points:
1. Did all of the hemicellulases, pectinase, and amylases show the compatible temperature and pH optima at 50 °C, pH 4.5? Please check it.
2. Why did it select the strain Klebsiella pneumoniae AP-407 for its ability of nitrogen fixation, HCN production, phosphate solubilization, potassium mobilization, siderophore production, ammonia, and IAA production?
3. Should it control the temperature of centrifugation?
4. It is suggested to give the definition of cellulases, xylanase, mannanase, amylases, and pectinase activities.
5. For the enzymatic hydrolysis of composite kitchen waste, should it conduct the pretreatment of shatter?
6. Table 1 is recommended to show the error data of total reducing sugars.
7. The abstract and conclusions should be improved with the data of this study.
8. Too many tables are shown. It is suggested to show several tables as figures.
9. The submission’s writing, organization, and expression of should be improved carefully. The manuscript is not well written. The language quality impairs reader understanding on several occasions. Additionally, the typescript of this text suffers from several problems. For example, the text “ml” could be “mL”.
Author Response

(The authors gave the same response as above.)

Reviewer 3 Report
The topic of the research article is of great interest. However, I would not recommend publishing the article in its current format as it requires lots of improvement. The main drawbacks of this manuscript
Below are several specific comments.
1. The English writing should be further improved, as there are many grammatical or typing errors. It is suggested to ask a native speaker to polish it.
2. there is a mix between American English and British English
3. minor comment:
Line 16 changes technique to technology
Line 61 changes its to their
Line 62 adds the before present
Line 65 removes a before separate
Line 91 changes poly-saccharides to polysaccharides
Line 92 is it carbohydrases or carbohydrates
Line 109 removes the before Hostel
Line 125 changes hydrolysing to hydrolyzing (( even American English or British English
Lines 151-154 changes hydrolysing to hydrolyzing (( even American English or British English
Line 193 changes equation to Equation
Line 195 adds a comma before and
Line 226 changes were to was
Line 261 adds a before novel
Line 329 changes biofertilzier to biofertilizer
Line 335 removes own
Line 338 changes biodegrdable to biodegradable
Line 338 changes biofertilzier to biofertilizer
Line 352 changes are to is
Line 360 changes process to professes
Line 365 changes were to was
Line 375 changes biofertilizer to biofertilizes
Line 423 changes biofertilizer to biofertilizes
Line 433 adds the before number
Line 437 changes (The aim of the current project was) to (The current project aimed)
Line 442 changes (Over a period of 75 days) to (Over 75 days) or (For 75 days)
Line 449 changes have to has
Line 454 adds the before root-rot
Line 469 changes biofertilizer to biofertilizes
Line 520 adds the before case
Line 532 changes are to is
Line 553 adds of before the soil
The topic of the research article is of great interest. However, I would not recommend publishing the article in its current format as it requires lots of improvement. The main drawbacks of this manuscript
Below are several specific comments.
1. The English writing should be further improved, as there are many grammatical or typing errors. It is suggested to ask a native speaker to polish it.
2. there is a mix between American English and British English
3. minor comment:
Line 16 changes technique to technology
Line 61 changes its to their
Line 62 adds the before present
Line 65 removes a before separate
Line 91 changes poly-saccharides to polysaccharides
Line 92 is it carbohydrases or carbohydrates
Line 109 removes the before Hostel
Line 125 changes hydrolysing to hydrolyzing (( even American English or British English
Lines 151-154 changes hydrolysing to hydrolyzing (( even American English or British English
Line 193 changes equation to Equation
Line 195 adds a comma before and
Line 226 changes were to was
Line 261 adds a before novel
Line 329 changes biofertilzier to biofertilizer
Line 335 removes own
Line 338 changes biodegrdable to biodegradable
Line 338 changes biofertilzier to biofertilizer
Line 352 changes are to is
Line 360 changes process to professes
Line 365 changes were to was
Line 375 changes biofertilizer to biofertilizes
Line 423 changes biofertilizer to biofertilizes
Line 433 adds the before number
Line 437 changes (The aim of the current project was) to (The current project aimed)
Line 442 changes (Over a period of 75 days) to (Over 75 days) or (For 75 days)
Line 449 changes have to has
Line 454 adds the before root-rot
Line 469 changes biofertilizer to biofertilizes
Line 520 adds the before case
Line 532 changes are to is
Line 553 adds of before the soil
Author Response

(The authors gave the same response as above.)

Reviewer 4 Report
1. How authors claimed that the bacterial strain (Klebsiella pneumoniae AP-407) used was non pathogenic? What is the source of this strain?
2. Briefly describe the enzyme essay procedures or their reference of estimation.
3. Statistical analysis is missing which seems that the data was not statistically analyzed.
4. Give some pictures of control and biofertilizer treated plants showing their effect on growth.
Author Response

(The authors gave the same response as above.)
